# Caregivers’ Perspective on the Psychological Burden of Living with Children Affected by Sickle Cell Disease in Kinshasa, the Democratic Republic of Congo

**DOI:** 10.3390/children10020261

**Published:** 2023-01-31

**Authors:** Patricia V. M. Lelo, Faustin Nd. Kitetele, Cathy E. Akele, David Lackland Sam, Michael J. Boivin, Esperance Kashala-Abotnes

**Affiliations:** 1Department of Infectious Diseases, Kalembelembe Pediatric Hospital, Kinshasa 012, Democratic Republic of the Congo; 2Centre for International Health (CIH), Faculty of Medicine, University of Bergen, 5020 Bergen, Norway; 3Department of Psychiatry, Michigan State University, East Lansing, MI 48824, USA

**Keywords:** sickle cell disease, perception, knowledge, caregivers, psychological burden, management, DR Congo

## Abstract

There is limited information on knowledge, perceptions, and management of sickle cell disease (SCD) in Africa in general and in the Democratic Republic of the Congo (DRC) in particular. This study explored knowledge, perceptions, and burden of 26 parents/caregivers of children with SCD in three selected hospitals in Kinshasa, DRC. We conducted a focus group with in-depth interviews with parents/caregivers of children affected with SCD. Four themes were discussed, including knowledge and perceptions, diagnosis and management, society’s perceptions, and the psychosocial burden and the quality of life of the family affected by SCD. The majority of participants/caregivers felt that society, in general, had negative perceptions of, attitudes toward, and knowledge about SCD. They reported that children with sickle cell are often marginalized, ignored, and excluded from society or school. They face a number of challenges related to care, management, financial difficulties, and a lack of psychological support. The results suggest the need to promote measures and strategies to improve knowledge and management of SCD in Kinshasa, DRC.

## 1. Introduction

Sickle cell disease (SCD) is a neglected, chronic, multisystem disorder of growing importance in the global health context [1,2,3]. It is caused by the inheritance of a mutation in the hemoglobin subunit Beta gene (HBB) that results in the production of a structurally abnormal form of β-globin. This variant form of hemoglobin, known as sickle hemoglobin or HbS, polymerizes reversibly under low oxygen tension to alter the shape and rheological properties of red blood cells, a phenomenon that is central to the pathophysiology of SCD [3,4]. Although SCD is most commonly caused by the homozygous inheritance of HbS, it can also result from the coinheritance of HbS with other mutations of the HBB gene. The most notable among them are a second structural hemoglobin variant, hemoglobin C (HbC), and β-thalassemia, a condition characterized by the reduced production of normal β-globin chains [3,4]. 

Genes causing hemoglobinopathies are found in about 5% of the world’s population. Each year, about 300,000 infants are born with major hemoglobin disorders—including more than 200,000 cases of SCD in Africa [5]. SCD is particularly common among people from sub-Saharan Africa (SSA), India, Saudi Arabia, and Mediterranean countries. In some parts of SSA, SCD affects up to 2% of newborns. More broadly, the prevalence of the sickle cell trait (healthy carriers that inherited the mutant gene from only one parent) reaches 10–40% in Equatorial Africa [5]. Babies born with SCD may not have symptoms for several months. When they do, symptoms include extreme tiredness or fussiness from anemia, painful swollen hands and feet, and jaundice. Babies may also have spleen damage that affects their immune system and increases their risk for bacterial infections. SCD symptoms typically start when babies are 5 to 6 months old. The signs and symptoms of medical conditions linked to SCD are anemia, stroke, splenetic sequestration, bacterial infections, and priapism. As people with SCD grow older, they may develop different and more serious medical problems that happen when organ tissues do not receive enough oxygen. As they grow older, most people with SCD have an increased risk of developing new medical conditions, some of which are life-threatening [6], such as Salter–Harris fractures that typically occur in older children in the growth plate when they have their growth spurt and when the physis is weak [7].

However, by being informed about their conditions and symptoms, people with SCD can learn to seek help at an early stage to improve management of their condition [6]. The sickle mutation has risen to high allele frequencies in many parts of Africa, India, and the Middle East because carriers (with Hb AS) are strongly protected against death from Plasmodium falciparum malaria [3,8,9]. Consequently, more than 90% of global SCD births—at least 5,280,000 births each year [10]—occur in resource-limited regions of the world. However, poor diagnostic facilities coupled with the low priority for SCD in the health plans of many countries in SSA mean that most children born with the condition in SSA are undiagnosed and die from preventable complications before their fifth birthday [11]. SCD is thus responsible for an increasing proportion of overall childhood mortality in SSA, reaching 6% or more in several countries within the region [12,13]. 

In Africa, the frequency of carriers of the sickle cell gene is variable and can reach a prevalence of 40% in some populations [14,15]. Cultural factors are particularly relevant to these problems because of beliefs and traditional practices [16]. People with SCD commonly report low self-esteem and feelings of hopelessness as a result of frequent pain, hospitalizations, and loss of schooling (in children) [16,17]. In the Democratic Republic of Congo, recent epidemiological data have shown that in the neonatal period, 2% of new-born are homozygous for the disease and about 40,000 births of sickle cell children are estimated each year. In the adult population, the carrying rate is 25%, and the homozygous form affects about 2% of individuals [18,19]. Although this figure is significant from an epidemiological point of view, the disease remains little known, resulting in high mortality in a country with limited resources [20,21]. 

The disease may impact on many aspects of the lives of patients, including education, employment, and psychosocial development due to recurrent pain and complications caused. Psychosocial issues for people with SCD and their families often result from the impact of chronic pain and symptoms on their daily quality of life, and society’s attitudes and perceptions toward people affected with SCD. Previous research in Africa revealed a negative attitude and perception of society toward SCD [16,22]. People had vague knowledge about SCD [23,24]. Other studies have shown that religious beliefs play a positive role in coping, including prayer, faith in God and doctors, and a hopeful approach to health challenges. Prayer and hope are commonly used as an effective coping strategy [25,26]. 

The DRC is the third country in the world and the second in Africa to bear the socioeconomic burden of SCD after India and Nigeria [27,28]. It has 20 to 30% carriers of the S trait (heterozygous AS), a birth rate of homozygous newborns (SS) of around 2%, and mortality of around 50 to 75% before the age of 5 years [27,29]. Considering the significant psychological and sociocultural burden that comes with this disease, we set out to explore the knowledge and perception of and attitudes toward SCD in Kinshasa, the capital of the DRC, a country highly affected by SCD.

## 2. Subjects and Methods

### 2.1. Study Design and Setting

We conducted a descriptive and qualitative study, using focus group discussions with parents/caregivers of children with SCD. We used the brainstorming approach, consisting of several open-ended questions to the parents/caregivers. These were then followed by the structured set of questions for the domains described below, to give them the opportunity to express themselves and thus facilitate the collection of information. 

The study was conducted in Kinshasa, the capital of the DRC, which comprises three referral hospitals specialized in the management of SCD: Kalembelembe Paediatric Hospital (KLPH) is a public and the only pediatric hospital in the country inaugurated in 1948 [30]. The KLPH is a health institution specializing in the management of children aged 0 to 15 years old, according to the standards of the World Health Organization (WHO) [31]. KLPH is specialized in the management of SCD and has other specialized pediatric services. Mabanga Hospital (Centre for Mixed Medicine and SS Anemia (CMMASS)) is the first hospital created in Central Africa and the only official institution in the DRC that specializes in the management of SCD [22,32]. The third health center selected was Monkole Hospital that is a private referral hospital created in 1991 and which collaborates with the ministry of health and comprises a specialized unit for the management of patients with SCD [22]. 

### 2.2. Sampling 

A total study sample of 26 parents/caregivers consented to participate in a focus group discussion. Of these, 24 were females, and most of the parents lived in Kinshasa and were workers. 

For the three hospitals described above, all parents and caregivers (non-biological parents primarily responsible for the care of the child) attending these hospitals with their child for sickle cell care were invited to participate in the focus group. 

In all, 36 (47%) of the 76 invited parents/guardians were from KLPH, 30 (40%) from CMMASS, and 10 (13%) from Monkole. Of all invited parents, only 26 responded positively to the invitation, representing 22 (84%) from KLPH, 3 (12%) CMMASS, and 1 (4%) Monkole. Overall, 50 (65%) of the 76 parents/guardians who declined to participate did so due to the COVID-19 pandemic. Each group included parents/guardians of children with SCD followed in these hospitals.

### 2.3. Data Collection

The focus group had two sessions, one of which had 12 participants and the other had 14 participants, and each session talked about the same issues with the same set of questions. The discussion was conducted in French (the official language) and Lingala (one of the national languages spoken in Kinshasa). It was guided by a physician facilitator and three observers. The first observer was a nurse who wrote participants’ responses in a logbook, and the second observer was the data manager who recorded the discussion with an audio recorder. The third observer was another physician who looked after SCD patients, who helped to answer questions from participants and facilitated the achievement of goals at the end of each session. All participants provided their informed consent to participate in the study and to record the discussion. 

To gather information, we used the brainstorming approach with parents and caregivers of children with SCD. While discussing, the questions were organized into 4 main themes, which were (1) knowledge and perceptions of SCD, (2) diagnosis and management of SCD, (3) society’s perception of children affected by SCD, and (4) the psychosociocultural burden of SCD and quality of life of family affected by SCD. We estimated the repetition of responses by respecting the basis of theoretical saturation (the point in data collection when new data no longer bring additional insights to the research questions) [33].

### 2.4. Data Analysis

Data analysis of approximately 22 hours 43 minutes of family story recordings was conducted using qualitative analysis of content based on themes and trends related to research questions. Deductive and inductive codes were developed and applied to the data. The Lingala interviews were transcribed verbatim into Lingala and translated into French, and the French interviews were transcribed verbatim directly into French. All interviews were verified by four team members and then combined with notes taken during the focus group and mini reports produced by the research team members. Data processing and analysis focused only on data related to the themes. The analysis was carried out in three main stages. As a first step, the transcripts were read several times to familiarize ourselves with the participants’ stories and to identify the themes associated with the relevant questions. All themes were saved and tagged with a unique code to compile a list of sub-categories for the aspects explored, and a MS Excel 2016 analysis matrix was used to organize the qualitative data. The analysis matrix presented online transcription data and in the themes/codes associated with the columns. Each interview transcript was represented by a row. In the second step, the researchers used the list of subcategories to code each interview transcript and used the analysis matrix to organize the data. In the third step, the sub-categories were merged into categories corresponding to the aspects explored, and the synthesis was made by searching for links, similarities, and differences. The principal investigator completed the analysis process and discussed it with other local authors and health partners. In addition, the use of working notes ensured the reliability of the analysis.

### 2.5. Ethics Approval and Consent to Participate 

Ethical clearance and permission to conduct this study were sought and obtained from the DRC National Health Ethics Committee at the Ministry of Health (code: 61/CNES/BN/PMMF/2018) and the Norwegian Regional Committee for Medical and Health Research Ethics (code: 2018/581-1) 

The purpose and nature of the study were explained to all participants. Informed consent was obtained from each participant.

## 3. Results

### 3.1. Sociodemographic Characteristics 

Demographic characteristics of the participants are presented in Table 1. Their age ranged between 34 and 38 years. All of them had children with SCD and were aged between 2 and 20 years old. Among caregivers, there was one couple, one grandmother, and one maternal aunt. The majority of the parents lived in Kinshasa and were workers. Regarding the medical history, two women reported a spontaneous abortion following birth of previous children with SCD. They believed the abortion was caused by stress and high blood pressure after announcing the SCD. One couple had lost a child at an early age due to SCD. Three women reported having divorced because of SCD. One woman reported that she refused to marry and chose to take care of her child herself in fear of being rejected by men because of her sick child. All parents acknowledged having their children followed up in one of the three selected centers offering management and care for people with SCD. Of all the parents who were invited to participate in the focus group but declined, the main reason was linked to the COVID-19 pandemic. Some declined because they found it difficult to talk about their children with SCD as it brings back sad memories. 

### 3.2. Themes

Below are presented responses by theme. Each participant was asked to give two types of responses, one on their own opinion and the other one on the general public view. 

#### 3.2.1. Knowledge and Perception on SCD

##### Knowledge

Most of the participants gave similar responses regarding knowledge and perceptions of SCD. Most parents declared having limited or vague knowledge of SCD before their marriage. Only two parents reported having clear knowledge about SCD before they married. 

Some of the responses to the question on the definition of SCD were as follows:


*“SCD means that the person suffers from SS anemia…”*

*Participant (P)1*



*“SCD child is the child who carries hemoglobin S in their blood…”*

*P2*



*“SCD is a genetic disease…” *

*P3*



*“Sickle cell disease means that the child has a problem in the blood, something is wrong in his/her blood, his/her blood does not circulate well, he/she has bone pain, so he/she has a problem in the blood. When someone is ignorant, the child is often transfused without parent realizing that the child may have SCD. My little daughter has been transfused four times, and we only knew later that it was SCD when we came here to Kinshasa. I thank God for that.”*

*P4*


##### Perceptions

Regarding the perceptions of disease, it was found that most parents think that SCD is a disease like any other. A quarter of parents (6.5 of 26) think like the community and believe that SCD is a curse or may originate from witchcraft. Parents were asked about attitudes toward and perceptions of SCD based on the perception of the general public and the school (Table 2).

Below are some of the responses given by parents/caregivers on their perception of SCD: 


*“I consider SCD as any other illness: For my little daughter, I give thanks to God and I leave everything in the hands of God.” *

*P4*



*“It is like any other disease; I take the example of tuberculosis if you follow the treatment well you will be healed. It is the same for this disease, if you follow the treatment well you will have peace.”*

*P5*



*“At first, I was very sore seeing the evolution of my child compared to other children, but now since my child is well followed-up, with a good treatment he will be healed, and he will be better.”*

*P6*



*“I will say that it is a disease like diabetes, HIV carriers, as they take their drugs, they live, just take the drugs well with a good follow-up, deepen some tests.”*

*P7*



*“At the family level we were talking about witchcraft, especially my parents-in-law, they said bad things, but now I’m used to it because I am already divorced, and the child is fine.”*

*P8*


#### 3.2.2. Diagnosis and Management of SCD

##### Diagnosis of SCD

The diagnosis of SCD is often confused with hemoglobin tests. Many of the participants were confused about the two tests, the blood type, and electrophoresis. The majority of participants did not know about SCD diagnosis before marrying. The summary regarding diagnosis and circumstances of disclosure is represented in Table 3. 


*“We didn’t know what SCD is, we had the confusion between blood type (hemoglobin) and electrophoresis.”*

*P9*



*“My older sister knew her status, but my brother-in-law, her husband, was ignorant. Before their wedding, our dad asked our brother-in-law to do the test and he brought the result of his blood type instead of electrophoresis.”*

*P3*



*“We had confusion between blood type and electrophoresis, we did the first and not the second.” *

*P10*


Some of the parents/caregivers reported being mis- or undiagnosed before receiving the proper test and correct diagnosis:


*“It’s really difficult, before the wedding we did this test but we didn’t know this kind of test is done in specialized hospitals, we went to a polyclinic or we were told that we were fine when our child was 7 years, he began to have health problems regularly, he was transfused three or four times, after that he was hospitalized and diagnosed SCD. We almost filed a complaint against the polyclinic in question, but it was no longer worth doing it because we were told that this type of test is done in a specialized hospital.”*

*P11*



*“We did the test at a non-specialized, inappropriate center, and the result was wrong, we got married believing that there was nothing, but in reality, there was a problem that we discovered along the way.” *

*P12*



*“I suspected something was wrong about my child, but after an examination in a local polyclinic the result of the test was not correct, but the health problem persisted and when we came here to Kalembelembe hospital, we took the test again and the test was positive.”*

*P13*


Circumstances of discovery of the disease

In most cases, the disease was discovered accidentally, during a medical consultation with symptoms such as fever, pallor, foot–hand syndrome, and joint pain or a hospitalization. In a few cases, friends or extended family members recognized the symptoms of SCD and encouraged parents to have their children tested.

Some participants said:


*“The first time, it was very difficult, we almost even separated, when our daughter was 2 years old, she was sick, we had given her antibiotics then her hands began to swell, the doctor asked us to do the electrophoresis test, and the result was positive, we were shocked we went to different local laboratories, the second test was positive too. The doctor who received us advised us to get separated, he told us if we continue together, we will have a second child with the same disease, he directed us to a hospital in the nearby because he does also work there. It felt like a pressure, we asked ourselves what can we do? After reflexing, we decided to continue together and didn’t get divorced. Finally, out of our five children, she is the only one with SCD. So from a scientific point of view, it is a disease that needs follow-up, but from a spiritual point of view, God heals every illness and that is our consolation.”*

*P5*



*“When my child was 7 years old, he started to have health problems regularly, he was transfused three or four times, and after he was hospitalized, we got to know that my child has SCD.”*

*P14*



*“At 9 months, my grandchild started to become ill, she was hospitalized, often transfused and one year after she was transfused again, we travelled to Kinshasa in 2019 and she was diagnosed with SCD. Now, I have faith my little girl will grow up and have children, I thank God.”*

*P4*



*“We did not know that our child was having SCD. She is now 14 years, she is the eldest twin, she had a young sister who started having symptoms at six years old, and she had symptoms such as anemia, hands-foot syndrome, and pain in her bones.”*

*P15*



*“We were ignorant about the condition and ended up losing the twin sister. Her twin sister died, but she has a growth problem, and she is fragile.”*

*P16*


Effect of the disclosure on parents

Fear, despair, and sadness dominate the picture when the result of SS anemia is disclosed to parents. One woman had high blood pressure due to stress. Society stigmatizes parents regarding the future of their children. The cultural context is that people think this disease is linked to witchcraft and that children with SCD will not reach adulthood and will die early.

Here are some of the answers from parents/caregivers of children with SCD:


*“According to society, children with SCD do not have a long life but for us parents, we will have to take care of them, we have the assurance that they will live longer and will be healthy.”*

*P17*



*“The first few days when we received the news, it was painful, but with advice from health workers, we accepted the situation and we found it bearable and “normal”. Unfortunately, because of SCD, my husband decided to leave me, and I am now alone with my child.”*

*P18*


Parents/caregivers’ opinion about disclosing the status of the child

Concerning the question about the disclosure to the child their SCD diagnosis, few parents had refused to inform the child. One parent said that the child was still very young, and others still did not want to stigmatize their children. On the other hand, several parents had announced the diagnosis to the child and believed that knowledge of the disease by the child would facilitate compliance with treatment, although they did not know how and when to do it. Some parents had a neutral position regarding disclosure to the child and prefer that health workers decide when to inform the child. 

They said:


*“To this day I have not yet informed my daughter, because I think she is still very young to be informed. Soon she will be 14 years old.”*

*P1*



*Laughter in the room……*



*“For me, as they are twins, and that her sister does not have this disease if I inform her, she will be the first to be embarrassed in front of her sister that is why I do not prefer to inform her, I cannot inform her.”*

*P15*



*“My child is aged 7 and she is informed.”*

*P19*



*“I told him, he knows, he takes the medicines, and he does not forget.”*

*P20*



*“Mine also knows, my little girl is 7 years old and has known for a long time, besides it is she who reminds me of the appointment with the doctor, the time to take the medicine.”*

*P21*



*“My niece is 8 months, she is not yet old to understand, even her brothers and sisters do not know, we discovered it only in August this year. We keep it a secret between me, her mother, and her father, the other children are still small, and if we tell them, they will spread the news to the neighborhood, they do not know yet how to keep the secret.”*

*P3*



*“If he starts to understand things I have to inform him because when he will grow up he will take himself medicines, he’s going to start asking questions, there are awake children! I have to feed him with hope, he cannot be uncomfortable with his brothers and sisters, and he must always be supported.”*

*P22*



*“My child is also 2 years old; I believe when he will begin school I will inform him, from 6 years.”*

*P23*


##### Management of SCD

All parents welcomed the evolution of medical management, and experiences of health services were mostly positive, as the majority of parents felt that medical staff have a good understanding and expertise of SCD. Most participants acknowledged that medical staff offer them good treatment. Their major preoccupation and concern were the cost and insufficient financial means to cover other aspects of care and management, such as the cost of paramedical care. Most of the time, parents felt discouraged because the family budget was disrupted by the increase in anemia crises of the child, which requires excessive expenses for support. All the participants declared that the care for SCD is very expensive, and it is not free of charge in public hospitals. Parents mentioned that it is desirable to continue with psychosocial support and management. They would like to be given a little more time during consultation and for their children to always be prioritized. To improve the management of SCD, parents especially proposed the involvement of the government in funding health care for children with SCD in particular, and all children in general:


*“My wish would be that the management of SCD becomes free of charge everywhere. When we bring children with SCD to the hospital, they are given priority.”*

*P23*



*“Also make the announcement to the benefactors for taking care of more patients because we all do not have the same financial means. ”*

*P2*



*“Our children need vaccine and follow-up, but both are very expensive, we do not have enough money.”*

*P24*



*“We hope that one day, the exams, the follow-ups and vaccines, all management for SCD will be free of charge.”*

*P25*


Parents’/caregivers’ views on the management of SCD

The parents of children with SCD proposed setting up self-support groups for sharing and discussing the problem of their children. The parents also advised supporting each other with encouraging words and showing empathy. 

These are some of the pieces of advice offered by some participants to encourage each other:


*“It is necessary to remove fear. the example of our couple, we knew before the wedding, there was even conflict in my family-in-laws, my mother-in-law is a nurse, my father knew that I was heterozygote AS, we even separated but love drew us and then, we got married, we had four children, but only one has SCD.”*

*P5*



*“For you who are divorced, and those who do not want to get in a new relation, I’ll advise you not to be afraid to meet another man who is homozygous AA. You can have a new life with a heterozygote man. We will advise you to ask your partner to get tested before and here (at Kalembelembe hospital) or at a specialized center. It is important. But if we meet a man who is a heterozygote, we will advise you not to get married to him and not follow the feeling if you are afraid of going into a similar situation, which can lead to divorce and having a child who is suffering with SCD.”*

*P7*


#### 3.2.3. Society’s Perception of Children Affected by SCD 

In the opinion of the participants, most community members believe that SCD is linked to or due to witchcraft. People with SCD commonly report low self-esteem, feelings of hopelessness as a result of frequent pain, repeated hospitalizations, loss of schooling, and stigmatization by society. 

##### Impact of SCD on Family of Children with SCD

Parents and siblings of children with SCD reported that they live in a state of constant stress. A mother told usthat she always brings her child with her everywhere because she is afraid to leave her with other people. She is mainly afraid that people will not manage to help her child if the child has strong pain due vasoocclusion. Having a sick child has impacted her live. The attention of the whole family is focused on the child with SCD. This special and regular attention given to the child sometimes causes jealousy among their siblings. Most parents reported that they accept the child without distinction although they live in constant fear. This is why religion has an important place in family life for acceptance of this situation and all that they endure. Most parents have faith in God regarding the proper evolution of their child’s health. They have faith in God and they pray and develop a strategy to accept their child’s illness. One of the women mentioned that she did not worry about all her pregnancies because she had faith in God. Out of her five children, only one had SCD. The impact of pain and symptoms on the children’s daily lives and negative society’s attitudes will affect families. For parents who already had a child with SCD, most moms reported that they were living under stress and did not have peace throughout the subsequent pregnancy to the extent that some of them had high blood pressure and two of them had an abortion.

The participants explained:


*“Knowing that I had a child with SCD child and another who died of this disease, when I knew that I was five months pregnant, I was so afraid and got high blood pressure, I ended up miscarrying my twins.”*

*P16*



*“I entered into fear when my child with SCD had a stroke, I also met different men who wanted to marry me, but when they see that I have a child with SCD, they leave. That’s why for the moment I prefer to stay alone with my child.”*

*P18*


The same question on the impact of SCD on family was asked to men, and one father said he would stay calm if his wife got pregnant again.

A male participant explained:


*“In my case I was the one who took the child to the test, when I came back knowing her status, I was able to handle the situation and I controlled her. If she gets pregnant again, I will not panic and will stay calm because we already have children with SCD, and I will know how to manage the situation. I will assume myself; we must follow the advice of the doctor.”*

*P26*


Most couples regretted being married with a partner who is heterozygote AS-AS. They said to themselves that if they had known, they would have given everyone the chance to marry an AA homozygote partner. Only one couple had agreed to marry each other knowing that they were both heterozygote AS-AS before marriage. Nevertheless, they said they did not realize what it is to live with a child having SCD. 

##### Impact of SCD on Schooling

The school is an environment of permanent stigmatization for children with SCD. The physiognomy of these children sometimes causes other children to marginalize, tease, and mock them at school, especially at a younger age. Once the child grows up, there is often overprotection from classmates.

Parents are worried because children with SCD often get sick and miss school and exams due to repeated hospitalization, which means that they are sometimes left behind compared to their classmates and are often frustrated. Many, in certain cases, drop out of school. One mother told us that her child will never go to school again because it is a waste of money because he never ends the year without getting sick. Discrimination from classmates, especially during childhood, may result in low self-esteem. Sometimes, teachers and school staff are not well informed about SCD and have difficulty understanding their challenges. Some teachers or staff consider children with SCD lazy and therefore want to keep them away from school activities. One father reported that his child with SCD was almost excluded from the school’s sports activities even though the school was informed about his condition. 

Some participants said:


*“I work at the school where my daughter is studying, and I don’t like the words teachers use to belittle sickle cell children.”*

*P1*



*“At school, when I compared his growth with his classmates, there is a big difference. His classmates are more developed than him. Sometimes he worries a lot about his growth retardation even though his friends care about him and show empathy and sympathy.”*

*P25*



*“The child knows his condition because he is taking the medication, but sometimes he feels frustrated because he does not do gymnastics at school. He is prevented from playing football and has a lot of restrictions. He always asks the same question: Mom why always me? My child feels alienated from the society.”*

*P8*



*“They are considered unfit, they will not reach the age of 30, but in the community, we see in reality many who are more than 40 years.”*

*P24*



*“Some students tease him, others are kind. He lives together with them, and they play together.”*

*P9*


There are some teachers who do not understand that they must take care of children who live with SCD because they cannot perform hard gymnastics. Some teachers force children to actively train or exercise. They give physical punishment to children with SCD. 

#### 3.2.4. Psychosociocultural Burden of SCD and Quality of Life of Family Affected by SCD 

The majority of participants think that society in general has negative attitudes toward and perceptions of people with SCD. A very large proportion of participants think that society does not have a good understanding of SCD and needs more awareness. If society was better informed, people would be more positive toward those with the condition. Quality of life in people with SCD is therefore more impaired than that of the general population. A family with SCD seems to be influenced by external factors such as religion, faith in God, superstitions, and stigma. Parents of children with SCD are often unwilling to hospitalize their children because of limited financial means. The community has a negative approach toward children and families affected by SCD. Many people in the community believe that children with SCD will not live long and will die at an early age. Parents think that there is limited knowledge and awareness of SCD in the community, and the disease is the desease is caused by witchcraft or due to bad luck. Children with SCD are often disregarded in society and are seen as a curse. Congolese cultural factors are particularly relevant to these problems because of beliefs and traditional practices. Congolese beliefs are often influenced by cultural and religious values, which also influence health behavior. Several parents reported that they often begin with traditional healing or religious healing (prayer) as a primary approach to treatment and end with medical treatment in case of failure. This may be due to the belief that the causes of illness are attributed to supernatural powers. There are cultural beliefs and understandings of the concepts of health, illness, causes, and treatment. The community beliefs may lead to negative perceptions and attitudes regarding SCD.

Below are the parents’ responses to the question of what society thinks about children with SCD:


*“Regarding growth, they are sometimes called sorcerer children, bewitched because they are shorter than same age children. My child had a problem with the spine, but after treatment, he is well.”*

*P10*



*“According to society, the SCD children are losers, do not have a long life but for us parents we will have to take care of our children because we have the assurance that the child will live and will be healthy.”*

*P14*



*“It is difficult to explain but it is considered as if they have a short life, there is no hope for them, and they will do nothing in life.”*

*P11*


## 4. Discussion

The goal of this paper was to assess the burden of SCD on parents/caregivers of children with SCD in Kinshasa, DRC, and to explore their perceptions and knowledge of the disease using a focus group approach with parents/caregivers of children affected by SCD.

Our findings suggest significant gaps in knowledge and perceptions of, attitudes toward, and practices with respect to SCD. The findings also suggest a heavy burden of the disease on the daily lives of families with children affected by SCD.

### 4.1. Knowledge and Perceptions of SCD

Our findings suggest that a large proportion of respondents believe that SCD is not a well-known disease in the society. The majority of participants did not have good knowledge of the disease, and they reported negative attitudes toward and perceptions of people with SCD. A previous study conducted in the DRC by Luboya and colleagues on the psychosocial impact of SCD on parents with children affected by SCD, and on the perception of SCD, suggested that the presence of SCD in the family is a source of conflict between couples [22]. Another study conducted in Nigeria by Kofi et al. explored the psychosocial impact of SCD in a Nigerian population, and its result shows that the majority of participants thought that society, in general, had a negative image of SCD through its attitudes and perceptions, although they did not feel that people had a negative approach towards them because of their SCD. They also reported that knowledge can also be influenced by external factors such as advice given by health workers, family support, culture, and professional authority [16]. Benoit Mbiya, in the DRC, found that 76% of participants knew about SCD, but without details. In another article by the same author, he found that 32% did not know about the disease [23,24]. In line with previous studies, there is acknowledgement of success in the management of SCD using a comprehensive approach focused on the patient and the family [34]. Caregivers in our studies have also acknowledged the importance of involving both the patients and caregivers for the successful management of SCD. 

Studies from the African continent suggest that living with a child with SCD is a source of rejection or curse in the family, and the child is often stigmatized [16]. Overall, our findings suggest poor perception and knowledge of this disease in Kinshasa in particular, and in some African countries due to a lack of information, such as those of previous studies. There is a need to promote sensitization and awareness among families and communities to reduce stigmatization. 

### 4.2. Diagnosis and Management of SCD

The majority of participants did not know that they were carriers of sickle cell genes before they married. Many of them indicated that it is not part of the culture to get tested for SCD prior to marriage, although some may know that the disease exists and is genetically transmitted. Instead, they ignore the disease until the first symptoms appear in children. Many parents/caregivers said they had faith in God and traditional medicine to heal them. A minority of parents who did not inform their young children about SCD did so to prevent them from growing up with worry and fear. This lack of information on the physiology of the disease, its genetic transmission, and its management would certainly be linked to the lack of awareness of parents and the community, as well as the tendency to use traditional medicine at first manifestations. These are consistent with previous studies [16,35]. We believe that parents’ reluctance to disclose the diagnosis is certainly related to poor knowledge of SCD. Parental denial and silence around the disease may suggest the devaluing and stigmatization of SCD in Congo in particular, and in Africa in general [22,36,37]. We believe that involvement at the national level through the National Sickle Cell Program in promoting screening prior to marriage would help to reduce the burden of the disease in the Congolese population.

There are currently significant improvements in the management of SCD, and most participants acknowledged progress that has been achieved in the management of people with SCD. Previous studies from the DRC have not reported improvement in the management of SCD, though most recent studies acknowledged improvement in private hospitals. They point out challenges due to lack of financial support in public hospitals, which makes access to care difficult for everybody. However, management of SCD in public hospitals remains a challenge as expenses for medical care are high and out of pocket. The government is not yet fully implicated in the cost coverage. Previous research has found similar challenges in the management of SCD [16,22]. The cost of health care is high (Every family will spend approximately USD 1000 to 2000/patient/year out of their pocket to pay for health care) [27]. Unlike private hospitals and hospitals collaborating with foreign partners where SCD is better managed, unfortunately, not all patients could attend these private hospitals. It is therefore important to establish a consensus with all parties to ensure equity in the management for all affected patients and their families. Most of the participants expressed their wishes to have the government more involved in sensitizing and mobilizing funds to improve the management of SCD in public hospitals. In line with previous studies, our finding suggests that the financial burden of managing SCD remains on parents as they have to pay medical fees out of pocket due to lack of social security, in addition to poverty. In addition, there is generally a lack of competence in the management of SCD, which leads to unnecessary expenses. We believe that to overcome this challenge, it is necessary to train more health professionals in the management of SCD and to create funded specialized centers that will alleviate the financial burden on parents/caregivers.

### 4.3. Society’s Perception of Children Affected by SCD 

Most of them reported a short life expectancy for children with SCD. Many believe that teachers need training and information regarding SCD as children do not receive sufficient help at school and are subject to mockery. Similar results have been reported in previous studies [15,16,22,37], such as teasing by colleagues due to jaundice and associated discoloration of their eyes. There were other major psychosocial issues experienced by children with SCD during their schooling, which were raised during the focus group. Parents/caregivers reported that uninformed teachers stigmatize children and prevent them from attending school activities, arguing that they are unfit or lazy. Children with SCD feel distressed and marginalized. This can arguably make some children develop low self-esteem and a negative perception of the school environment. We believed that this type of perception and behavior in the school environment is due to the lack of knowledge and information about the disease among teachers. However, it is worth emphasizing that some children with SCD perform relatively well at school when having supportive parents who encourage them to go to school and keep them as physically active as possible. Children without supportive families often cope inadequately, which leads to secluded lives. Previous research has also shown that the rate of school maladjustment among children with SCD can cause them to drop out of school [10,13,16]. 

One of the reasons that was given by parents/caregivers to keep their children with SCD at home and away from school was that most of them fear early death. There is a need to improve knowledge and awareness of SCD among communities and work toward an appropriate curriculum to facilitate adaptation to school for children with SCD.

### 4.4. Psychosociocultural Burden of SCD and the Quality of Life of the Family Affected by SCD

All parents/caregivers who participated in the focus group reported that people in general will have negative attitudes and perceptions towards children with SCD. Almost everybody reported some stigma from the community toward children with SCD and their families. This is not new. Indeed, previous research from Mali has reported similar findings suggesting that poor psychological adjustment of children with SCD has remained relatively constant over time [26,38]. Congolese and numerous other African communities rely on traditional treatment, and religious healing (prayer) as an alternative approach to treatment in addition to standard treatment, which is in many cases the first choice of care. The reason given for choosing traditional treatment as the first choice is that people believe that the disease, more than being chronic, is attributed to “divine punishment”, supernatural causes, superstitions, and witchcraft. In addition, the majority of participants believe that traditional treatment is more effective. There is a national strategic plan for the management of SCD in the DRC [39], but many parents reported that these guidelines and strategies are not really followed to improve the care and management of people with SCD. The lack of free care in public hospitals has been also reported in other studies [36]. In Africa, cultural factors are particularly relevant to these problems because of beliefs and traditional practices. Other studies have shown that religious beliefs have an important place in families with SCD [16,26,37,40,41,42]. The finding from our study suggested that apart from financial difficulties, which is a huge burden, many parents reported relying on religion or prayers as it gives them strength to face the social, cultural, and financial challenges of living with SCD. Many parents/caregivers are determined to accompany their children on this long journey of chronicity through prayers. On the other hand, other parents consider SCD a burden and choose to go away from their families. This is especially among fathers who abandon their children to their mothers and start a new life away from their families. 

In urban areas, parents/caregivers instead choose religion as the first choice compared to traditional medicine. This is a new trend with the establishment of new churches and mostly in cases of financial deprivation to cover the cost of medical care. 

### 4.5. Strengths and Limitations of the Study

Our study has strengths in that it investigates the burden of SCD from parents/caregivers’ perspective in Kinshasa. Parents experienced deep pain while expressing themselves and results may be affected by their emotion; however, the study provides useful information as perceived by affected families. The study highlights the extent to which families affected by SCD are facing challenges in terms of sociocultural, psychological, and financial difficulties in their daily life to care for children with SCD. The study also discovered a huge gap in knowledge and management of SCD both at community level as well as among health professionals.

The study provides a basis for a better understanding of the situation and highlights the need to develop and implement appropriate interventions with emphasis on education to improve awareness in the community, reduce stigma, and provide appropriate management in terms of psychological and financial support. Our findings suggest that there is a need to act at two levels, the educational level to improve awareness and the medical level to better manage SCD and reduce incidence, morbidity, and mortality related to SCD. This is in line with the WHO’s priorities to establish primary health care as a coordinating center for person-centered care [21] as suggested in other African studies about SCD. 

The study has some limitations. The main one is that the study was conducted during the COVID-19 pandemic in 2020, and as a result, several parents or caregivers declined to participate in the focus group for fear of catching COVID-19 in the hospital setting because the focus group was held at KLPH. Participants were also afraid to take public transport which involved being in proximity to other passengers and increased the risk of contracting COVID-19. The cost of public transport in the city and districts was high because the number of passengers was reduced in public transport. Therefore parents who were able to attend the focus groups were those who could afford to pay extra money to take public transport, as ticket price had increased. Most of the participants lived far from KLPH, especially those invited from Monkole and Mabanga. All these factors resulted in fewer participants responding to the invitation. There was gender inequality because men were under-represented, and our outcomes are affected by this, with our findings mostly representing women’s perspectives. However, we believe the results are useful, as it is usually mothers who look after families in the DRC and in many African cultures [22]. Another limitation is related to respondents’ varying levels of education. Different levels of education may have had an impact on our outcomes, as the knowledge, perceptions, and management of SCD may vary from group to group. However, we found that more than 80% of respondents had a higher level of education than high school. The number of participants was not representative as we used only parents/caregivers of children with SCD as respondents in this study. Our results cannot therefore be extrapolated to the general population of the DRC. However, the objective of our focus group was to investigate knowledge and perceptions of the parents’/caregivers’ perspective in Kinshasa. A study in remote DRC areas would have probably produced different results. Nevertheless, the study contributes to our understanding of the psychosocial impact of SCD in the Congolese context and from the parents’/caregivers’ perspective.

## 5. Conclusions

Caregivers/parents of children with SCD perceived SCD as a sociocultural, psychological, and financial burden. There is a need to increase awareness among the population to reduce stigmatization. Even though participants acknowledged efforts and improvements made by the government in the management of SCD in the DRC, there is a need for access to free-of-charge medical care, especially for families with SCD as out-of-pocket payments for medical care constitute a huge financial burden. There is a need for psychosocial interventions to support families with SCD to alleviate their psychological burden. A study at a larger scale is needed to better understand the burden of chronic disease in the DRC. 

## Figures and Tables

**Table 1 children-10-00261-t001:** Demographic characteristics of the parents/caregivers of children with SCD in Kinshasa, DRC.

Variables	N = 26
Age Groups:	
24–28	1 (3.85%)
29–33	8 (30.77%)
34–38	3 (50%)
39–43	3 (11.54%)
44–56	2 (7.69%)
Gender:	
Male	2 (7.69%)
Female	24 (92.31%)
Residence:	
Urban area	21(80.77%)
Rural area	5(19.23%)
Participant level of education:	
Primary	3(11.54%)
Secondary	15(57.69%)
University	7(26.92%)
None	1(3.85%)
Occupation:	
Employed	4(15.38%)
Student	1(3.85%)
Housework	11(42.31%)
Health worker	1(3.85%)
Self-employment	10(38.46%)

**Table 2 children-10-00261-t002:** The proportion of different responses to attitudes toward and perception of SCD as perceived by the general public and the school as reported by parents in Kinshasa, DRC.

	Yes
Category	N = 26
General public:	
Negative Attitudes and perception	26
Negative Approach by People	24
Stigmatization	20
Lack of awareness	19
Superstitions and witchcraft	23
SCD child does not have a long life	23
Traditional practices	24
Influenced by cultural values	19
Influenced by religious values	26
School:	
Teachers need knowledge	21
Stigmatization/Discrimination	19
Teasing/Mockery	18

**Table 3 children-10-00261-t003:** Proportion of responses given by parents/caregivers regarding the diagnosis of SCD and the circumstances of disclosure.

	Yes
Diagnosis of SCD	N = 26
1. Parents’ knowledge of the diagnosis of their children:	
Number of misdiagnosed children	23
Good understanding of the diagnose	2
Bad understanding of the diagnosis	24
2. Number of parents with knowledge of the diagnosis before the wedding:	
Knowledge	2
No knowledge	24
3. Circumstances of discovery of the disease:	
Accidentally discovered	21
At birth after the child was screened	5
4. Parents/caregivers’ response regarding disclosure to their children:	
Refuse to disclose	7
Agree to disclose	14
To be unsure about it	5

## Data Availability

The datasets generated for this study are available on request due to ethical restrictions.

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
