# Peer review of "Caregivers’ Perspective on the Psychological Burden of Living with Children Affected by Sickle Cell Disease in Kinshasa, the Democratic Republic of Congo"

_children, 2023, doi:10.3390/children10020261_

Round 1

Reviewer 1 Report

Very interesting work, aiming to evaluate the burden of the disease for parents and families of child with SCD, the perception and knowledge of society on the disease, the integration of children in school, the strategies developed by parents and caregivers. The article is pleasant to read, alternating quotes from the participants and comments from the authors, who formulate suggestions for a better management of SCD both at the educational level and the health level (national program with financial support, training of health professionals, opening of funded specialized centers,sensitizition and awareness campaigns, premarital screening....

Reviewer 2 Report

Overall submission:

-       Aligns with the aim of this Special issue as it focuses on “‘hard-to-reach’ families due to rurality, ethnicity, poverty and culture” and bring international experience in family centered intervention.

Reviewer 3 Report

Thanks for sharing such a well written and informative study. However, I made few suggestions to further improve the general knowledge of the study.

1. In line 51: There should be a citation at the end of the symptoms of SCD.

2. In line 53: Could the authors state the possible new medical conditions that SCD children might develop? I will suggest citing this study based in Africa (Nigeria)which is endemic for SCD; Opara, N.U.; Osuala, E.C.; Nwagbara, U.I. Management of Salter–Harris Type 1 Fracture Complicated with Osteomyelitis in a Sickle Cell Disease Patient: A Case Report and Review of Literature. Medicines 20229, 50. https://doi.org/10.3390/medicines9100050

3. 
